# Effect of Green Tea on Weight Gain and Semen Quality of Rabbit Males

**DOI:** 10.3390/vetsci9070321

**Published:** 2022-06-26

**Authors:** Andrej Baláži, Alexander V. Sirotkin, Pavol Makovický, Ľubica Chrastinová, Alexander Makarevich, Peter Chrenek

**Affiliations:** 1Institute of Farm Animal Genetics and Reproduction, NPPC, Research Institute for Animal Production in Nitra, Hlohovecká 2, 951 41 Lužianky, Slovakia; lubica.chrastinova@nppc.sk (Ľ.C.); alexander.makarevic@nppc.sk (A.M.); 2Department of Zoology and Anthropology, Constantine the Philosopher University in Nitra, Nábrežie Mládeže 91, 949 01 Nitra, Slovakia; asirotkin@ukf.sk; 3Department of Biology, Faculty of Education, J. Selye University, Bratislava 3322, 945 01 Komárno, Slovakia; pavel.makovicky@gmail.cz; 4Institute of Biotechnology, Faculty of Biotechnology and Food Science, Slovak University of Agriculture in Nitra, Tr. A. Hlinku 2, 949 76 Nitra, Slovakia

**Keywords:** rabbit male, sperm concentration and motility, testicular histology, weight gain

## Abstract

**Simple Summary:**

The consumption of green tea can reduce the occurrence of inflammatory and cardiovascular diseases, diabetes, cancer and obesity in humans. Due to its antioxidant and anti-microbial effects, green tea is considered an additive to feed in animal production to substitute for antibiotics. The goal of this study was to evaluate the action of the green tea plant (*Camellia sinensis*, L) given in powder form on male rabbits’ reproductive functions (sperm concentration and motility, testicular morphology) and some non-reproductive traits (weight gain, blood metabolic/haematological and biochemical parameters). The obtained results demonstrate a reduction in rabbit weight gain, but deterioration of sperm quality under the influence of green tea feeding. This study suggests that green tea can affect the metabolic and reproductive systems of rabbit males in different manners, which should be taken into account when assessing the impact of green tea on a living organism.

**Abstract:**

The goal of the current study was to evaluate the action of the green tea plant (*Camellia sinensis*, L) on male rabbit reproduction and some non-reproductive indexes. Male rabbits were fed either a standard diet (control group) or a diet enriched with green tea powder (experimental groups; E): 5 g (E1) or 20 g (E2) per 100 kg of the milled complete feed mixture. Weight gain, sperm concentration, total and progressive motility, as well as haematological, and biochemical parameters and changes in testicular tissue histomorphology were evaluated. Feeding with green tea, at both tested concentrations, decreased weight gain per week and the total average weight gain compared to the control group (*p* < 0.05). Furthermore, green tea decreased sperm concentration, motility and progressive motility in the group fed with a lower dose (5 g) of green tea powder (*p* < 0.05), whilst a higher dose (20 g) was neutral. Some haematological and biochemical indexes, like medium-size cell count (MID), mean corpuscular haemoglobin concentration (MCHC), platelet percentage (PCT), levels of phosphorus (P) and total proteins (TP) were decreased in one or both experimental groups (*p* < 0.05), whilst the triglyceride level (TG) was increased in the E2 group (*p* < 0.05). The thicknesses of the testicular seminiferous tubules and epithelial layer were not affected by any concentration of green tea powder (*p >* 0.05). These observations suggest that green tea in the diet may have an adverse effect on rabbit growth and sperm quality, but their effect may be potentially dose-dependent.

## 1. Introduction

Plants are an excellent natural source of antioxidants. Green tea expresses antioxidative, pro-apoptotic, anti-microbial, endocrine, metabolic and angiogenic actions. Because of these properties, the consumption of green tea can decrease the occurrence of inflammatory and cardiovascular diseases, diabetes, cancer and obesity [1,2,3,4,5]. Because of its antioxidant and anti-microbial effects, green tea has been accepted as a feed additive in agricultural animal production as a natural adaptogen and substitute for antibiotics [6,7]. Green tea has been successfully used as a feed admixture in fish [8], chickens [9,10,11], calves [12], dairy cows [13] and pigs [14]. Some clinical and epidemiological studies have shown the health benefits of epigallocatechin-gallate (EGCG) in the prevention of diabetes and obesity [15] via modulations of energy balance, food intake, endocrine systems, and lipid and carbohydrate metabolism [16,17]. On the other hand, Baláži et al. [18] observed some adverse effects of dietary green tea on rabbit doe’s ovarian functions, fecundity and viability, perhaps because of changes in ovarian steroid hormone release, cell apoptosis and blockade of the large antral ovarian follicles ovulation. In contrast to females, green tea can positively affect male reproductive processes. Several studies have documented that herb antioxidants in a semen extender were associated with the improvement of spermatozoa quality. The addition of plant antioxidants into a semen extender resulted in the improvement of bovine [19], canine [20], avian [21], mouse [22] and rat [23] semen quality. The effect of green tea on semen parameters seems to be due to its antioxidative properties. The consumption of green tea by male rats increased sperm concentration and improved their viability [24]. Feeding rabbits with green tea at a dose of 6 g green tea/kg diet increased their semen volume, sperm concentration, motility, pH and decreased sperm abnormalities [25]. Furthermore, green tea extract was able to prevent the suppressive effect of para-nonylphenol, an environmental contaminant, on rat sperm number, motility, and viability as well as, the diameter of seminiferous tubules, the thickness of the germinal epithelium, the total volume of the testis, the volume of seminiferous tubules and testis weight [26]. This extract also prevented the effect of some environmental contaminant such as deltamethrin, on the number of rat spermatogenesis progenitor cells and spermatozoa, and their morphology and motility, as well as epithelial height, the diameter of the seminiferous tubules and testosterone release [27]. On the other hand, there is evidence that green tea or its constituents can not only improve but also decrease sperm quality. De Amicis et al. [28] reported that EGCG, added at low doses (2 µM and 20 µM), improved human sperm motility and viability, but when given at a high concentration (60 µM), it decreased these sperm parameters. Similarly, Kročkova and Kovačik [29] reported increased sperm motility in rabbits administered with 0.75 mg/L of green tea extract, while the motility was decreased at higher concentrations of green tea (1.5–3.0 mg/L). Chandra et al. [30] observed a negative dose-dependent action of green tea extract (1.25–5 g%) on rat sperm counts, spermatogenesis (number of spermatids) and testosterone production. 

The rabbit is a domestic animal species with a significant economic impact. Rabbits have a short life cycle, a short gravidity period, are very prolific, and have a high feed conversion capacity. Furthermore, rabbits are becoming increasingly popular as a source of healthy meat with high nutritional value. The influence of green tea on the male rabbit reproductive system (testicular morphology and quality of semen), however, has not been investigated yet. The present study was designed to investigate the efficacy of different dietary levels of green tea, as a source of water-soluble antioxidants, on the male rabbit reproductive system. 

The general aim of our experiments was to examine the action of green tea, given in the form of powder, on male rabbits’ reproduction functions (sperm concentration and motility, testicular morphology), as well as on some non-reproductive parameters (weight gain, blood metabolic/haematological and biochemical parameters).

## 2. Materials and Methods

### 2.1. Animals

Clinically healthy rabbit males of the New Zealand White line (National Agricultural and Food Centre-NAFC Nitra, Slovak Republic) were used in this experiment. The animals were housed in individual cages under a constant photoperiod of 16 h of light and 8 h of darkness during the research. Humidity and temperature in the hall were recorded continuously by means of a thermograph positioned at the same level as the cages (the average relative temperature and humidity during the year were 17 ± 3 °C and 60 ± 5%). Male rabbits (*n* = 30), 45 days old, were divided into three groups: control (C; *n* = 10) and two experimental groups (E1; *n* = 10 and E2; *n* = 10). The males in the control group were fed with a commercially available complete feed mixture (Tekro, s.r.o., Nitra, Slovakia). Green tea powder (right powdery green tea of Chinese origin, supplied by Oxalis, Prague, Czech Republic) was added at two distinct doses (E1: 5 g; E2: 20 g) to 100 kg of the milled complete feed mixture and after that mixed again (6-mm pellets). The males were fed with standard food (C) or food supplemented with green tea (E1, E2) for 90 days up to the achievement of sexual maturity (135 days), and they were weighed each week. Average weekly and total weight gains from 45 to 135 days of age were recorded. The males in the control group (C) were still fed with standard food, and the males in the experimental groups (E1, E2) were still fed with food supplemented with green tea powder for an additional 55 days so that the whole experiment lasted for 190 days. Semen collection was performed between 135 and 190 days of the experiment.

### 2.2. Semen Collection and Handling

Semen was collected from sexually mature rabbit males using a pre-heated artificial vagina once a week in a regular manner [31]. From each male, seven collections of semen samples were done. The semen was transported to the laboratory in a water bath at 37 °C and processed individually for an in vitro evaluation. Only ejaculates with a white colour were used in the experiments. Samples containing urine and cell debris were discarded, whereas gel plugs were removed. Immediately after collection, fresh rabbit spermatozoa were diluted in a saline solution (0.9% NaCl; Braun, Germany) at a ratio of 1:8 (v:v), placed into Leja Standard Count Analysis Chamber (depth of 20 microns) (MiniTüb, Tiefenbach, Germany) and evaluated underneath a Zeiss AxioScope A1 microscope using the CASA software (Sperm VisionTM; MiniTübe, Tiefenbach, Germany) according to the instructions of the manufacturer. For each sample, six microscopic view fields were investigated for average concentration (CON; 1 × 10^9^), percentage of total motility (TM; motility > 5 μm s^−1^) and progressively motile spermatozoa (PM; motility > 20 μm s^−1^).

### 2.3. Haematological and Biochemical Parameters

At the end of the experimental process (after 190 days of green tea permanent feeding), blood samples were collected from the *arteria carotis communis* of each rabbit male before sacrificing into tubes without anticoagulant (blood serum for biochemical analysis was separated by centrifugation at 1200× *g* for 20 min at laboratory temperature) and into EDTA-treated tubes (whole blood for haematological analysis). Many biochemical and haematological parameters were measured, as reported in Table 1 and Table 2. Methods are described in detail in our previous study [18].

### 2.4. Histological Processing, Analysis and Morphometry of Testes

Immediately after animal slaughtering, one testis per animal was fixed in a 4% formalin solution (Sigma-Aldrich Corp., St. Louis, MO, USA) for 24 h. Subsequently, the testes were treated by standard histological methods using an automated tissue processor, a Leica ASP6025 (Leica Microsystems, Wetzlar, Germany), and then stored in paraffin blocks using a Leica EG 1150H paraffin-embedding station (Leica Microsystems). Slices ranging from 3 to 5 μm in thickness were cut from each sample using a microtome Leica RM2255 (Leica Microsystems) and embedded on standard glass slides (Bamed s.r.o, Litvínovice, Czech Republic). The first slice set was stained with a haematoxylin-eosin stain (DiaPath, S.p.A., Martinengo, Italy) (Figure 1A–C). The second slice set, for the verification of nucleic acids for better nuclei morphology highlighting, was stained using a Toluidine Blue Polychromatic (DiaPath, S.p.A.) (Figure 1D–F). The last set of sections, for the verification of reticulum and collagen type I, was stained using a Sirius red kit (DiaPath, S.p.A.) (Figure 1G–I). The prepared samples were assessed using a Nikon Eclipse E600 microscope (Nikon Corporation Instruments Company, Tokyo, Japan). For an objective evaluation, a histomicrometry of the seminiferous tubules was performed. In each sample, the diameters of 100 tubules and the thickness of 100 epithelial layers were evaluated using the NIS-Elements version 3.0 software (Laboratory Imaging, Ltd., Prague, Czech Republic).

### 2.5. Statistical Analysis

The experiments were carried out on 30 animals (C = 10; E1 = 10; E2 = 10) monitored during the period of growth, fattening and sexual maturity. For the determination of data normality distribution, the Shapiro–Wilk test was used. Afterwards, obtained data were assessed by a t-test and one-way ANOVA test using SigmaPlot 11 (Systat Software Inc., Erkharth, Germany). All values are expressed as means ± standard error of the mean (SEM) or in percent. Differences from control at *p* < 0.05 were accepted as statistically significant.

## 3. Results

### 3.1. Effect of Green Tea on Weight Gains and Semen Quality

The effect of different concentrations of green tea plant powder added to the diet on the average weekly and total weight gains of the male rabbits was evaluated in our study. The highest average weight gain of rabbit males per week in the control group (C) was 183.08 ± 15.7 g, while in the experimental groups, it was 154.2 ± 12.9 g and 149.3 ± 14.6 g for the E1 and E2 groups, respectively (Figure 2). Likewise, the total average weight gain was the highest in the control group (C; 2380.3 ± 133.9 g) compared to both experimental groups (E1; 2005.67 ± 86.1 g and E2; 1925.7 ± 67.1 g; Figure 3).

In our study, the sperm ejaculates collected from the experimental and control males were analysed for their quality. Sperm concentration, total motility and progressive motility were decreased in the group receiving a lower dose of green tea (E1; Figure 4, Figure 5 and Figure 6), whilst in the E2 group, there was a tendency toward decreased progressive motility.

### 3.2. Effect of Green Tea on the Testis Morphometry, Blood Biochemical and Haematological Indexes

Macro- and micrometry of the testes did not uncover the impact of feeding with green tea on the seminiferous tubules or thickness of epithelia. No significant influence of green tea added at any dose on these parameters was observed, although a slight tendency to shrink the seminiferous tubules and epithelial layer in the E2 group occurred (Figure 7 and Figure 8).

The numbers of animals involved in weighting during the experiments (*n* = 10 per each group) were different compared with the number of animals left at the end of the experiment when blood samples were collected and analysed for haematological and biochemical markers (C—*n* = 7; E1—*n* = 7; E2—*n* = 5). We assume that these differences in measured markers are probably not caused by the green tea action but rather by individual variations and the small number of animals in tested groups. No significant green tea-induced alterations in key metabolic/haematological parameters were found (Table 1), but in both experimental groups, medium-size cell count (MID), and in the E2 group, mean corpuscular haemoglobin concentration (MCHC) and platelet percentage (PCT), were decreased. Regarding biochemical parameters, levels of phosphorus (P) and total proteins (TP) were decreased in experimental groups. On the contrary, the level of triglycerides (TG) was increased in E2 group (Table 2).

## 4. Discussion

### 4.1. Green Tea Can Suppress Weight Gains

In our present study, green tea extract, added at both doses, reduced body weight in male rabbits. Many studies have illustrated the effects of green tea on metabolism and body weight [5,17,32]. Green tea and its most represented constituent, EGCG, can decrease weight gain in rats [33,34,35,36,37], mice [38], rabbits [39] and humans [40,41]. The reduction could be due to increased lipolysis because green tea can decrease body fat accumulation, triglycerides, serum cholesterol, hepatic cholesterol absorption, and cholesterol bile acid levels to alter the processing of plasma lipoproteins, triglyceride biosynthesis and lipoprotein secretion. This effect of green tea was observed in rats [34], mice [42,43], rabbits [44], and humans [32,40,41], but not in cows [13]. The weight-reducing effect of green tea was reported in relatively small animals (mice, rats, rabbits) and humans, but not in large animals, like cows [13], which can be explained, for example, either by the different response of large animals to green tea preparation or by the improper concentration of green tea preparation in rations of cows, which is not sufficient to bring any effect on body weight. Furthermore, green tea was able to obviate obesity in mice [38]. In addition, tea caffeine can increase energy expenditure and, therefore, reduce fat reserves [5].

Green tea is a well-known antioxidant [45,46,47] and, therefore, green-tea-induced weight loss may also be explained by its antioxidant action. However, we did not investigate the antioxidant properties of green tea on rabbits, which is a limitation of this study. An alternative hypothesis explaining green tea-induced fat and weight loss was formulated by Rothenberg et al. [48]. It was expected that the weight-loss effectiveness of green tea is on account of carbohydrate digestive enzyme inhibition and subsequent reactions of undigested carbohydrates with gut microbiota, which forms short-chain fatty acids, which contrarily enhance lipid metabolism. The third hypothesis explains the tea-induced weight loss by its capability to suppress cell proliferation and support their apoptosis. At the least, such an influence of EGCG on cancer cells [49] and rabbit healthy ovarian cells [18] has been reported. This effect could explain the ability of EGCG to reduce the offspring’s growth rate and slightly increase pup loss in rats [50] or the increase in the number of liveborn and weaned pups and female mortality in rabbits fed by green tea [18]. 

Some previous studies did not uncover any dietary effect of green tea on body weight in rats [51], humans [32] and rabbit does [18] (in contrast to rabbit bucks in the present experiments). For that reason, green tea’s impact on body weight and metabolism could be species-dependent. From a practical point of view, our research suggests that green tea is not useful as a biostimulator of rabbit meat production and growth.

### 4.2. Sperm Concentration and Motility

In our experiment, feeding with green tea decreased the quality of rabbit male ejaculate. Rabbits fed with a lower dose (5 g/100 kg) of green tea had decreased sperm concentration, motility and progressive motility. There was a tendency to decrease progressive motility in the group with a higher dose (20 g/100 kg) of green tea. These observations do not correspond to the previous reports concerning the positive action of dietary green tea on sperm concentration and quality in rats [24] and rabbits [25]. On the other hand, they are in line with the previous observation on the effect of feeding with green tea on rat spermatogenesis and sperm count [30]. The differences in the action of tea on sperm observed by different authors could be due to variations in the initial reproductive state of males or dose-dependent differences in tea effects. De Amicis et al. [28] and Kročková and Kováčik [29] demonstrated the opposite action of green tea extract or EGCG added at low and high doses on human and rabbit sperm quality. Furthermore, the adverse effect of dietary green tea on rabbit sperm can be explained by a non-specific action of green tea on growth, proliferation, apoptosis and viability of cells and the whole animals, as mentioned above. This hypothesis is supported by the effect of dietary green tea on rabbit growth, viability and female reproduction observed in our previous studies [18]. On the other hand, these changes are probably not to be explained by tea’s action on metabolism because no substantial effect of dietary green tea on rabbit metabolic parameters was observed in the present experiments (see Table 2). Thus, the present data suggest that dietary green tea can have an adverse influence on rabbit sperm, which cannot be beneficial for rabbit male reproduction. 

Both the present and previous studies demonstrated the dose-dependent effect of green tea on rabbit semen quality and growth. In some studies, green tea has been shown to worsen the semen quality when given at low doses, while in other studies, it does this at high doses. The differences in the response of semen to this additive could be due to different sensitivities or responsibilities of semen used in different experiments with green tea. Identification of the tea constituents responsible for this effect, as well as mechanisms of their action on sperm, require further studies. Nevertheless, it might be hypothesized that green tea used in different experiments may have different contents of biologically active molecule(s) affecting sperm quality. It also cannot be excluded that the sperm used in different experiments had different abilities to receive and respond to green tea constituents.

### 4.3. Effect of Green Tea on Testicular Histology

It might be expected that the changes in sperm functions and quality observed after feeding rabbits with green tea can be induced by changes in the state of spermatogenic testicular tissue. Nevertheless, the histomicrometry of the testes performed in our studies did not reveal any impact of feeding with green tea on the seminiferous tubules or thickness of epithelia. There was only a slight tendency to shrink the seminiferous tubules and epithelial layer in both experimental groups, which probably did not cause reduced semen quality.

These observations differ from the observations of other authors on green tea’s influence on testicular tissue in rats. In particular, dietary green tea increased the diameter of seminiferous tubules, the number of spermatogonia, thickness of the germinal layer, and Sertoli and Leydig cells [23] and protected testicular tissue against oxidative damage [52], acrylamide [53] and cadmium [54]. Therefore, the lack of association between sperm quality and histomorphometrical parameters of testis indicates that the adverse effect of green tea on sperm is not due to the changes in testicular morphology. Nevertheless, it does not exclude green tea’s influence on functional testicular parameters, e.g., production of sufficient motile sperm, which requires further elucidation.

### 4.4. Effect of Green Tea on Haematological and Biochemical Parameters

The action of green tea on blood cells is less inspected, while the existing data are contradictory and insufficient. Cyboran et al. [55] did not uncover green tea’s impact on the porcine erythrocyte membrane. Hu et al. [56] reported that EGCG protected from a radiation-induced reduction in white blood cell count, red blood cell count and haemoglobin in mice blood. Green tea catechol decreased mice platelet aggregation [57]. El-Ratel et al. [39] informed that feeding rabbits with green tea increased haematocrit, haemoglobin concentration, mean corpuscular volume (MCV), and total white blood cell count (WBC), and on the other hand, decreased white platelet count (WPC). Our research did not uncover an important impact of dietary green tea on the majority of rabbit haematological and biochemical parameters except for a reduction in medium-size cell count (MID) in the E1 group, and, in the E2 group, the mean corpuscular haemoglobin concentration (MCHC) and platelet percentage (PCT). Regarding biochemical parameters, levels of phosphorus (P) were decreased in the E1 group, and total proteins (TP) were decreased in both experimental groups. On the contrary, the level of triglycerides (TG) was increased in the E2 group. Reduced medium-size cell count, mean corpuscular haemoglobin concentration, and platelet percentage could trigger the inhibition of platelet restoration and bone marrow erythropoiesis [58,59] and the resulting protective and transport blood functions. We can assume that the observed changes in some metabolic and haematological parameters compared to control, as investigated in our experiments, could be associated with reduced weight gain and male semen quality. However, since the numbers of animals involved in weighing during the experiments (*n* = 10 per each group) were different compared with the number of animals at the end of the experiment when blood samples were collected and analysed for haematological and biochemical markers (C—*n* = 7; E1—*n* = 7; E2—*n* = 5), we might suggest that these differences in measured markers cannot be caused by the action of green tea, but rather by individual variations and the small number of animals in tested groups. 

The treatment of aged rats with GTE caused a significant increase in levels of total protein, albumin, globulin, albumin/globulin (A/G) ratio, blood haemoglobin (Hb), red blood cells (RBCs), white blood cells (WBCs), platelet counts and levels of liver and kidney reduced glutathione (GSH). Additionally, a significant decrease in the serum activities of aspartate aminotransferase (AST), alanine aminotransferase (ALT), alkaline phosphatase (ALP) and liver and kidney levels of lipid peroxidation product, malondialdehyde (MDA) and a highly significant decrease in levels of total lipids, total cholesterol and triglycerides was observed compared to control rats [60].

## 5. Conclusions

Our experiments demonstrate that green tea reduced weight gain both at lower and higher doses, while it exerted a negative effect on sperm quality when given at a lower dose. These results suggest that green tea can affect the metabolic and reproductive systems of male rabbits in different manners. Therefore, in future research, possible conclusions about its effect on animal organisms should be taken with care. 

## Figures and Tables

**Figure 1 vetsci-09-00321-f001:**
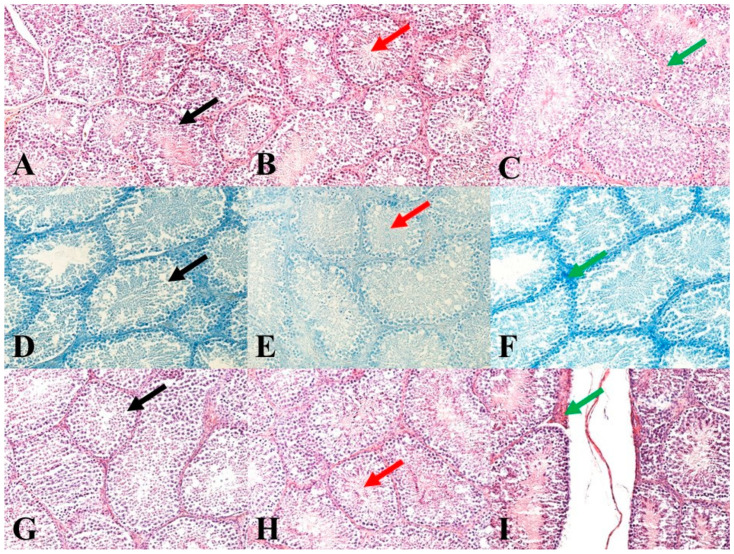
Representative images of testicular histology. (**A**): Control group. HE: 200×. (**B**): E1 group. HE: 200×. (**C**): E2 group. HE: 200×. (**D**): Control group. Toluidine Blue Polychromatic: 200×. (**E**): E1 group. Toluidine Blue Polychromatic: 200×. (**F**): E2 group. Toluidine Blue Polychromatic: 200×. (**G**): Control group. Sirius red: 200×. (**H**): E1 group. Sirius red: 200×. (**I**): E2 group. Sirius red: 200×. Black arrows show seminiferous tubules and germ cells in various stages of development. Red arrows show mature sperm in seminiferous tubules. Green arrows show septa between seminiferous tubules.

**Figure 2 vetsci-09-00321-f002:**
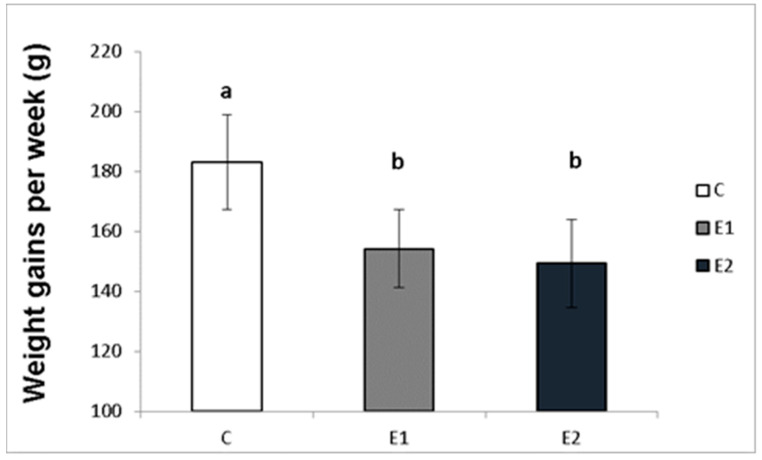
Average week weight gains (g) of male rabbits fed with green tea plant supplemented into complete feed mixture. C (without dietary green tea). E1—5 g green tea/100 kg normal food. E2—20 g green tea/100 kg normal food. Values and error bars express means ± SEM. a vs. b—difference is significant at *p* < 0.05.

**Figure 3 vetsci-09-00321-f003:**
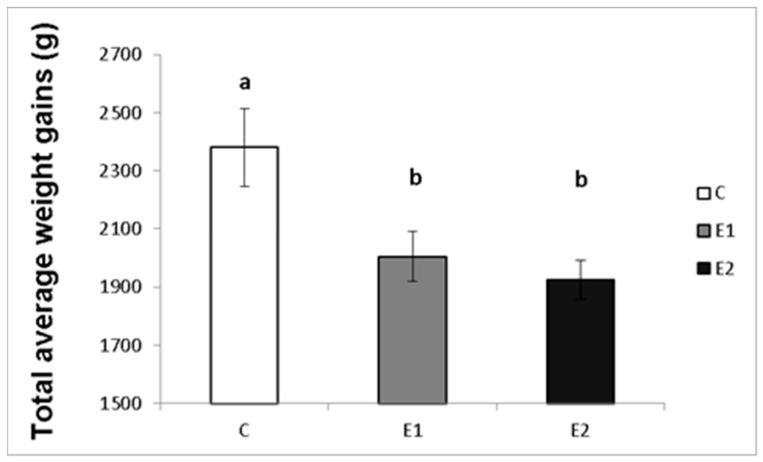
Total average weight gains (g) of male rabbits fed with green tea plant supplemented into complete feed mixture. C (without dietary green tea). E1—5 g green tea/100 kg normal food. E2—20 g green tea/100 kg normal food. Values and error bars express means ± SEM. a vs. b—difference is significant at *p* < 0.05.

**Figure 4 vetsci-09-00321-f004:**
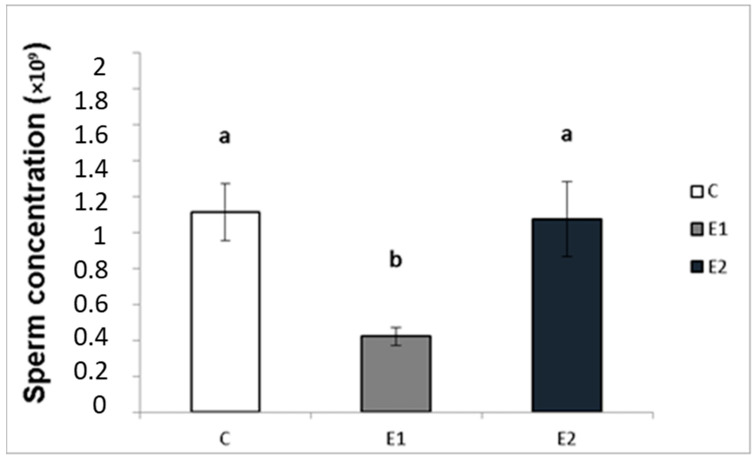
Sperm concentration in rabbits fed with green tea. C (without dietary green tea). E1—5 g green tea/100 kg normal food. E2—20 g green tea/100 kg normal food. Values and error bars express means ± SEM. a vs. b—difference is significant at *p* < 0.05.

**Figure 5 vetsci-09-00321-f005:**
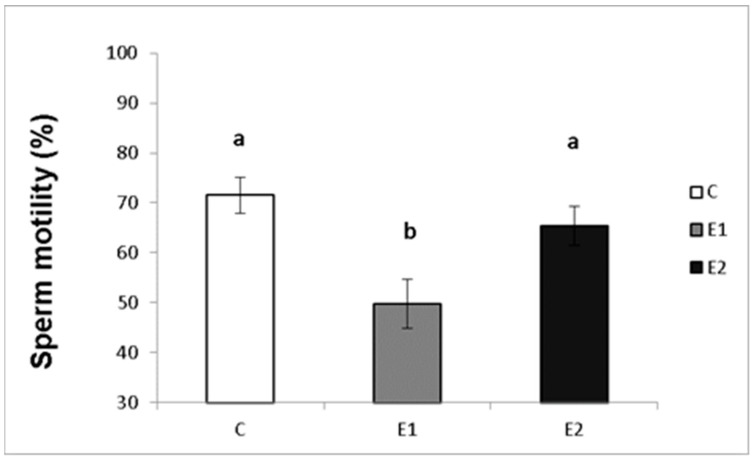
Sperm total motility in rabbits fed with green tea. C (without dietary green tea). E1—5 g green tea/100 kg normal food. E2—20 g green tea/100 kg normal food. Values and error bars express means ± SEM. a vs. b—difference is significant at *p* < 0.05.

**Figure 6 vetsci-09-00321-f006:**
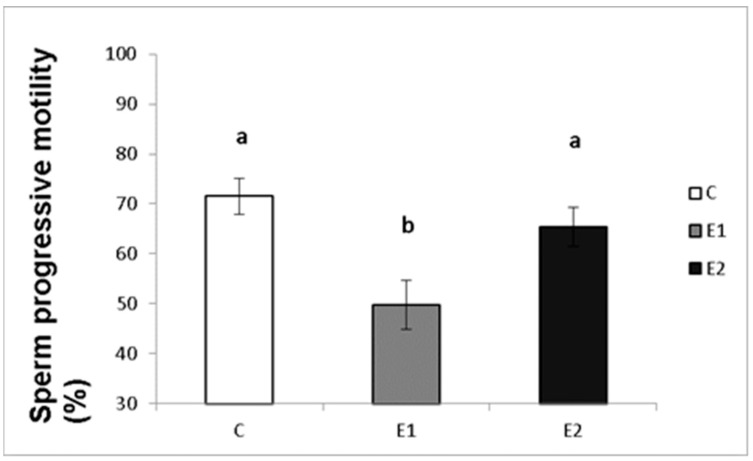
Sperm progressive motility in rabbits fed with green tea. C (without dietary green tea). E1—5 g green tea/100 kg normal food. E2—20 g green tea/100 kg normal food. Values and error bars express means ± SEM. a vs. b—difference is significant at *p* < 0.05.

**Figure 7 vetsci-09-00321-f007:**
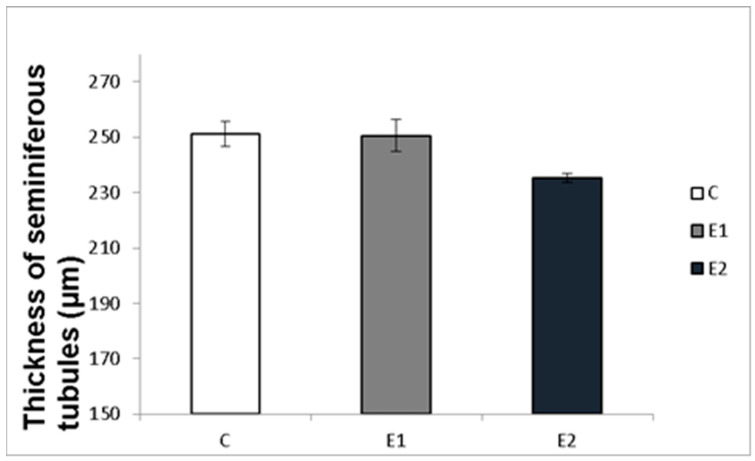
Effect of green tea on testicular seminiferous tubules. C (without dietary green tea). E1—5 g green tea/100 kg normal food. E2—20 g green tea/100 kg normal food. Values and error bars express means ± SEM.

**Figure 8 vetsci-09-00321-f008:**
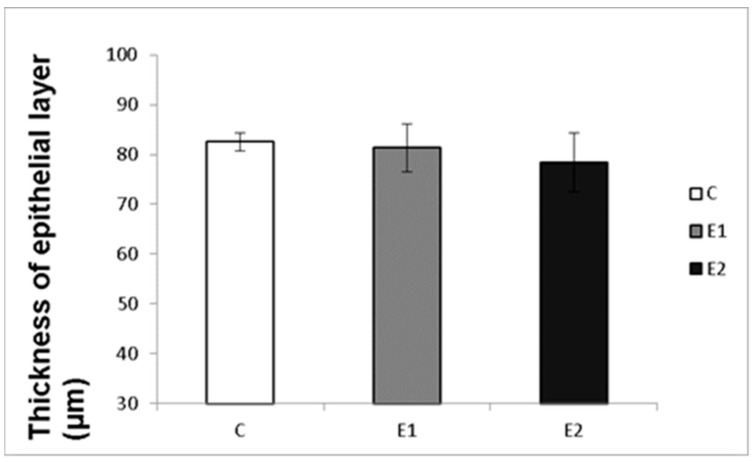
Effect of green tea on the thickness of tubular epithelial layer. C (without dietary green tea). E1—5 g green tea/100 kg normal food. E2—20 g green tea/100 kg normal food. Values and error bars express means ± SEM.

**Table 1 vetsci-09-00321-t001:** Effect of green tea on haematological parameters of rabbit blood.

Haematological Parameter	C	E1	E2
LYM (10^9^.L^−1^)	3.54 ± 1.40	3.41 ± 0.56	4.24 ± 2.08
WBC (109.L^−1^)	8.31 ± 0.97	6.36 ± 0.74	6.64 ± 1.32
GRA (10^9^.L^−1^)	4.36 ± 1.32	2.73 ± 0.93	2.14 ± 0.24
MID (10^9^.L^−1^)	0.41 ± 0.04 ^a^	0.21 ± 0.06 ^b^	0.26 ± 0.11 ^b^
MI (%)	5.15 ± 0.84	3.25 ± 0.91	3.77 ± 1.7
LY (%)	44.93 ± 9.58	56.98 ± 9.81	61.9 ± 5.86
GR (%)	49.9 ± 9.93	39.80 ± 9.44	34.3 ± 7.09
HGB (g.L^−1^)	143.00 ± 5.34	135.00 ± 3.76	131.33 ± 2.60
RBC (10^12^.L^−1^)	6.03 ± 0.25	5.60 ± 0.13	5.68 ± 0.09
MCV (fl)	53.00 ± 0.00	53.50 ± 0.87	53.67 ± 0.67
HCT (%)	32.03 ± 1.34	29.92 ± 1.07	30.41 ± 0.38
MCHC (g.L^−1^)	447.50 ± 2.33 ^a^	451.75 ± 5.22 ^a^	432.33 ± 4.49 ^b^
MCH (pg)	23.78 ± 0.14	24.13 ± 0.32	23.17 ± 0.13
PLT (10^9^.L^−1^)	244.00 ± 51.33	238.50 ± 51.10	138.68 ± 35.03
RDWc (%)	17.80 ± 0.31	18.48 ± 0.71	19.87 ± 0.22
MPV (fl)	6.55 ± 0.35	6.10 ± 0.15	6.07 ± 0.27
PCT (%)	0.16 ± 0.03 ^a^	0.15 ± 0.03 ^a^	0.08 ± 0.02 ^b^
PDWc (%)	31.80 ± 1.23	29.48 ± 0.57	30.23 ± 1.36

C (without dietary green tea). E1—5 g green tea/100 kg normal food. E2—20 g green tea/100 kg normal food. LYM—lymphocytes count; WBC—total white blood cell count; GRA—granulocytes count; MID—medium-size cell count; MI—medium-size cell percentage; LY—lymphocyte percentage; GR—granulocytes percentage; HGB—haemoglobin; RBC—red blood cell count; MCV—mean corpuscular volume; HCT—haematocrit; MCHC—mean corpuscular haemoglobin concentration; MCH—mean corpuscular haemoglobin; PLT—platelet count; RDWc—red cell distribution width; MPV—mean platelet volume; PCT—platelet percentage; PDWc—platelet distribution width. ^a,b^ Values within a row with diverse superscripts differ significantly at *p* < 0.05.

**Table 2 vetsci-09-00321-t002:** Effect of green tea on biochemical parameters in rabbit blood.

Biochemical Parameter	C	E1	E2
Ca (mM.L^−1^)	2.97 ± 0.08	2.83 ± 0.05	2.98 ± 0.17
P (mM.L^−1^)	1.52 ± 0.06 ^a^	1.38 ± 0.01 ^b^	1.37 ± 0.06 ^b^
Mg (mM.L^−1^)	1.26 ± 0.06	1.27 ± 0.03	1.16 ± 0.13
Na (mM.L^−1^)	149.08 ± 0.38	147.10 ± 0.57	149.43 ± 1.16
K (mM.L^−1^)	6.82 ± 0.15	6.60 ± 0.202	6.48 ± 0.25
Cl (mM.L^−1^)	110.15 ± 0.95	108.20 ± 1.21	108.13 ± 0.41
Urea (mM.L^−1^)	7.77 ± 0.67	6.41 ± 0.55	7.26 ± 0.83
Glu (mM.L^−1^)	6.69 ± 0.18	6.77 ± 0.20	6.51 ± 0.06
TP (g.L^−1^)	72.64 ± 1.80 ^a^	65.60 ± 2.66 ^b^	67.14 ± 2.27 ^b^
ALT (µkat.L^−1^)	0.90 ± 0.04	0.95 ± 0.06	1.00 ± 0.12
AST (µkat.L^−1^)	0.62 ± 0.07	0.65 ± 0.05	0.70 ± 0.09
ALP (µkat.L^−1^)	0.14 ± 0.03	0.18 ± 0.13	0.12 ± 0.07
GGT (µkat.L^−1^)	0.06 ± 0.04	0.05 ± 0.02	0.07 ± 0.06
CK (µkat.L^−1^)	37.96 ± 3.17	34.96 ± 3.47	41.68 ± 2.33
Cholesterol (mM.L^−1^)	0.75 ± 0.11	1.07 ± 0.28	1.02 ± 0.18
Bilirubin (µM.L^−1^)	5.69 ± 0.59	5.89 ± 0.71	5.98 ± 0.72
TG (mM.L^−1^)	0.43 ± 0.04 ^a^	0.55 ± 0.16	0.59 ± 0.07 ^b^

C (without dietary of green tea). E1—5 g green tea/100 kg food. E2—20 g green tea/100 kg normal food. Glu—glucose; TP—total proteins; ALT—alanine amino transaminase; AST—aspartate aminotransferase; ALP—alkaline phosphatase; GGT—gamma glutamyl transferase; CK—creatine kinase; TG—triglycerides. ^a,b^ Values within a row with diverse superscripts differ significantly at *p* < 0.05.

## Data Availability

The data presented in this study are available in the article.

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
