# Peer review of "Effect of Green Tea on Weight Gain and Semen Quality of Rabbit Males"

_vetsci, 2022, doi:10.3390/vetsci9070321_

Round 1

Reviewer 1 Report

The paper under review deals with the evaluation of the effect of green tea, given in a form of powder, on rabbit male reproduction functions (sperm concentration and motility, testicular morphology), as well as on some nonreproductive indexes (weight gain,
blood metabolic/haematological and biochemical indexes).
The general topic of the investigation and methodology are in the scope of the Veterinary Sciences.

The study showed that green tea added at dose 5 g per 100 kg feed had an adverse effect on rabbit growth and sperm quality, whilst a higher dose (20 g per 100 kg feed) was neutral.

Remarks

It is surprising that a low dose of green tea had a negative effect on semen quality, while a higher dose did not. Previous studies have shown that lower doses of green tea improved semen quality, while higher doses reduced it. Please discuss this discrepancy in more depth.

Conclusions should be based on the results obtained. The authors did not study the effects of green tea on semen quality in other animal species or humans, hence the conclusions should be re-written.

Author Response

Response to Reviewer 1 Comments

Dear rewiever

Thank you for your good and inspirating remarks.

Point 1: It is surprising that a low dose of green tea had a negative effect on semen quality, while a higher dose did not. Previous studies have shown that lower doses of green tea improved semen quality, while higher doses reduced it. Please discuss this discrepancy in more depth.

Response 1: Both the present and previous studies demonstrated the dose-dependent effect of green tea on rabbit semen quality and growth, whilst in some studies tea reduced semen quality when given at low doses, whilst in other studies it do it at high dose. The differences in the response of semen to additive could be due to different sensitivity or responsibility of semen used in different experiments to green tea. Identification the tea constituents responsible for this effect, as well as mechanisms of their action on sperm require further studies. Nevertheless, it might be hypothesized,

that tea used in different experiments has different contents of biological active molecule(s) affecting sperm quality and/or sperm used in different experiments had different ability to reception and response to tea constituents, which resulted different response of this sperm to tea administration. These explanations have been included into the revised version of manuscript.

Point 2: Conclusions should be based on the results obtained. The authors did not study the effects of green tea on semen quality in other animal species or humans, hence the conclusions should be re-written.

Response 2: We accepted the reviewers' comment and the conclusion was completely changed.

Thank you once again for your time to improve the quality of our manuscript.

Reviewer 2 Report

Dear authors,

Thank you for your interesting work 'Green tea can suppress weight gain and semen quality in rabbit males'. I would suggest to use a more general type of  title. I would like you to enhance and compare the other species effect of green tea to rabbits. Please change the conclusions, you should refer to mainly to your results than others.

Author Response

Response to Reviewer 2 Comments

Dear rewiever

Thank you for your good and inspirating remarks.

Point 1: I would suggest to use a more general type of title.

Response 1: There was applied a more general type of title. “Effect of green tea on weight gain and semen quality of rabbit males”.

Point 2: I would like you to enhance and compare the other species effect of green tea to rabbits.

Response 2: To discussion was added text: The weight-reducing effect of Green tea was reported in relatively small animals (mice, rat, rabbit) and human, but not in large animals, like cows [13], which can be explained, for example, either by different response of large animals to green tea preparation, or by improper concentration of green tea preparation in rations of cows, which is not sufficient to bring any effect on body weight.

Point 3: Please change the conclusions, you should refer to mainly to your results than others.

Response 3: We accepted the reviewers' comment and the conclusion was completely changed.

Thank you once again for your time to improve the quality of our manuscript.

Reviewer 3 Report

The study provides new useful information on the effect of Green tea diet supplementation in rabbit males. The manuscript presents a well-performed experimental design, investigating different mechanism that could underlying the effects of Green tea.

However, an English revision is mandatory and there are some major issues that should be improved for the final approval.

Some recent references are missing, please add them.

Introduction

Line 33: Please change in “An excellent natural sources of antioxidants are plants”

Lines 47-48: Please change with something like “that the addition of plant antioxidants in a semen extender were associated with the improvement of bovine [19], canine [20], avian [21], mouse [22] and rat [23] semen quality. The effect of green tea on semen parameters seems to be due to their anti‐oxidative properties.”

Lines 57-58: Please change with “This extract prevented also the effect of some environmental contaminant such as deltamethrin….”.

Lines 64-66: Please rephrase.

Line 69: Please add a sentence or two to specify why “Rabbit is a domestic animal species with significant economical impact.”

Materials and methods

Lines 100-101: It is not clear when the semen collection when performed during green tea administration, from 135 days to 190? Please clarify it.

Lines 119-120: Please change with “A lot of biochemical and haematological parameters were measured as reported in Tables 1 and 2.”

Lines 157-163: Please add the test used to verify distribution of data.

Results

Line 187: Please change with “ sperm concentration, total and progressive motility”.

Line 188: Please change in “ group receiving a lower dose of green tea “

Tables: Please specify the superscripts of E2 group for MID and P or E1 group for MCHC, PCT and TG.

Discussion

Lines 273-275: Please discuss more why Green tea has controversial effects in different gender and species.

Lines 277-297: Please add some phrase on the possible antioxidant effect of green tea, and add also that a limitation of this study is that this effect was not investigated.

Lines 338-342 since the numbers of animals involved in weighting during the experiments (n = 10 per 338 each group) were different compared with number of animals at the end of the experi 339 ment, when blood samples were collected and analysed for haematological and biochem 340 ical markers (C n= 7; E1 n= 7; E2 n= 5), we might suggest that these differences in 341 measured markers cannot be caused by the green tea action, but rather by individual var 342 iations and small number of animals in tested groups. “ Please add this part also in the material and methods or results.

Conclusions

Lines 355- 357: Please change in something like “From practical point of view, it suggests that green tea can be suitable for the improvement of performance and fertility in some species including humans, but controversial results are reported in rabbit with also a decreasing of weight gains and sperm quality.”.

Author Response

Response to Reviewer 3 Comments

Dear rewiever

Thank you for your good and inspirating remarks.

Point 1: Some recent references are missing, please add them.

Response 1: There were added 3 new references.

  1. Reto, M.; Almeida, C.; Rocha, J.; Sepodes, B.; Figueira, M.E. Green Tea (Camellia sinensis): Hypocholesterolemic effects in humans and anti-inflammatory effects in animals. Food and Nutrition Sciences 2014, 5, 2185-2194. doi: 10.4236/fns.2014.522231.
  2. Peluso, I.; Serafini, M. Antioxidants from black and green tea: from dietary modulation of oxidative stress to pharmacological mechanisms. British Journal of Pharmacology 2017, 174:1195e208. https:// doi.org/10.1111/bph.13649.
  3. Prasanth, M.I.; Sivamaruthi, B.S.; Chaiyasut, C.; Tencomnao, T. A review of the role of green tea (Camellia sinensis) in an-tiphotoaging, stress resistance, neuroprotection, and autophagy. Nutrients 2019, 11, 474. https://doi.org/10.3390/nu11020474.

Point 2: Line 33: Please change in “An excellent natural sources of antioxidants are plants”

Response 2: We accepted the reviewers' comment and the sentence was rewritten.

Point 3: Lines 47-48: Please change with something like “that the addition of plant antioxidants in a semen extender were associated with the improvement of bovine [19], canine [20], avian [21], mouse [22] and rat [23] semen quality. The effect of green tea on semen parameters seems to be due to their anti‐oxidative properties.”

Response 3: We accepted the reviewers' comment and the sentence was rewritten.

Point 4: Lines 57-58: Please change with “This extract prevented also the effect of some environmental contaminant such as deltamethrin….”.

Response 4: We accepted the reviewers' comment and the sentence was rewritten.

Point 5: Lines 64-66: Please rephrase.

Response 5: We accepted the reviewers' comment and the sentence was rephrased as: Similarly, Kročkova and Kovačik [29] reported increased sperm motility in rabbits admin-istered with 0.75 mg/L of green tea extract, while the motility was decreased at higher con-centrations of green tea (1.5-3.0 mg/L).

Point : Line 69: Please add a sentence or two to specify why “Rabbit is a domestic animal species with significant economical impact.”

Response 6: There were add sentences: It has a short life cycle, a short gestation period, it is notably very prolific, and it has a high feed conversion capacity. Furthemore, rabbit is becoming increasingly popular as a source of healthy meat with high nutritional value.

Point 7: Lines 100-101: It is not clear when the semen collection when performed during green tea administration, from 135 days to 190? Please clarify it.

Response 7: Semen collection is performed between 135 and 190 days of the experiment. It was added to material and methods.

Point 8: Lines 119-120: Please change with “A lot of biochemical and haematological parameters were measured as reported in Tables 1 and 2.”

Response 8: We accepted the reviewers' comment and the sentence was rewritten.

Point 9: Lines 157-163: Please add the test used to verify distribution of data.

Response 9: For the determination of data normality distribution, the Shapiro–Wilk test was used.

Point 10: Line 187: Please change with “ sperm concentration, total and progressive motility”.

Response 10: The sentence was corrected.

Point 11: Line 188: Please change in “ group receiving a lower dose of green tea “

Response 11: The sentence was corrected.

Point 12: Tables: Please specify the superscripts of E2 group for MID and P or E1 group for MCHC, PCT and TG.

Response 12: The superscripts for MID, P, MCHC, PCT were specified. Statistical differences for TG in E1 group is not signifficant, so there is not superscript.

Point 13: Lines 273-275: Please discuss more why Green tea has controversial effects in different gender and species.

Response 13: To discussion was added text: The weight-reducing effect of Green tea was reported in relatively small animals (mice, rat, rabbit) and human, but not in large animals, like cows [13], which can be explained, for example, either by different response of large animals to green tea preparation, or by improper concentration of green tea preparation in rations of cows, which is not sufficient to bring any effect on body weight.

No more references about gender was found, so phrase “gender-dependent” has been removed from the text.

Point 14: Lines 277-297: Please add some phrase on the possible antioxidant effect of green tea, and add also that a limitation of this study is that this effect was not investigated.

Response 14: Green tea is a well-known antioxidant [45-47] and, therefore, green-tea induced weight loss may be also explained by its antioxidant action. However, we did not investigate an antioxidant properties of green tea on rabbits, which is a limitation of this study.

Point 15: Lines 338-342 “since the numbers of animals involved in weighting during the experiments (n = 10 per each group) were different compared with number of animals at the end of the experiment, when blood samples were collected and analysed for haematological and biochemical markers (C ‐ n= 7; E1 ‐ n= 7; E2 ‐ n= 5), we might suggest that these differences in measured markers cannot be caused by the green tea action, but rather by individual variations and small number of animals in tested groups. “ Please add this part also in the material and methods or results.

Response 15: Under standard husbandry conditions, animal numbers are also reduced by natural mortality, therefore, the male´s number after 190 days duration of experiment was lower. This part of your comment was also added to section “Results”.

Point 16: Lines 355- 357: Please change in something like “From practical point of view, it suggests that green tea can be suitable for the improvement of performance and fertility in some species including humans, but controversial results are reported in rabbit with also a decreasing of weight gains and sperm quality”.

Response 16: The conclusion was completely rewritten according to the proposal of other 2 rewievers, because conclusions should be based on the results obtained. The authors did not study the effects of green tea on semen quality in other animal species or humans.

Thank you once again for your time to improve the quality of our manuscript.

Round 2

Reviewer 3 Report

Lines 354-357: Please change in : "A clear comprehension of the efficient mechanisms of green tea effect on rabbit male reproduction requires further studies and analysis of other physiological parameters. From practi al point of view, it suggests that green tea can be suitable for the improvement of performance and fertility in some species including humans, but not in rabbits, where it does not improve and even decrease weight gains and sperm quality. "

Author Response

Dear rewiever

Thank you for your good and inspirating remarks.

Point 1: Lines 354-357: Please change in : "A clear comprehension of the efficient mechanisms of green tea effect on rabbit male reproduction requires further studies and analysis of other physiological parameters. From practical point of view, it suggests that green tea can be suitable for the improvement of performance and fertility in some species including humans, but not in rabbits, where it does not improve and even decrease weight gains and sperm quality. "

Response 1: You thought probably the chapter conclusion, because the line numbers have changed in the last corrected version. The conclusion was completely rewritten according to the proposal of other 2 rewievers, because conclusions should be based on the results obtained. New conclusion is: “Our experiments demonstrate that green tea reduced weight gain both at lower and higher doses, while it exerted negative effect on sperm quality, when given at lower dose. These results suggest that green tea can affect metabolic and reproductive systems of rab-bit males in different manners. Therefore, in future research, possible conclusions about its effect on animal organisms should be taken with care”.

Thank you once again for your time to improve the quality of our manuscript.
